# Soybean-Derived Peptides Attenuate Hyperlipidemia by Regulating Trans-Intestinal Cholesterol Excretion and Bile Acid Synthesis

**DOI:** 10.3390/nu14010095

**Published:** 2021-12-27

**Authors:** Haksoo Lee, Eunguk Shin, Hyunkoo Kang, HyeSook Youn, BuHyun Youn

**Affiliations:** 1Department of Integrated Biological Science, Pusan National University, Busan 46241, Korea; hs950303@gmail.com (H.L.); egshin94@gmail.com (E.S.); kanghk94@gmail.com (H.K.); 2Department of Integrative Bioscience and Biotechnology, Sejong University, Seoul 05006, Korea; 3Department of Biological Sciences, Pusan National University, Busan 46241, Korea

**Keywords:** transintestinal cholesterol excretion, soybean, hyperlipidemia, bioactive peptide

## Abstract

Increased triglyceride, cholesterol, and low-density lipoprotein (LDL) levels cause hyperlipidemia. Despite the availability of statin-based drugs to reduce LDL levels, additional effective treatments for reducing blood lipid concentrations are required. Herein, soybean hydrolysate prepared via peptic and tryptic hydrolysis promoted trans-intestinal cholesterol excretion (TICE) by increasing ATP-binding cassette subfamily G member 5 (ABCG5) and ABCG8 expression. The peptide sequence capable of promoting TICE was determined via HPLC and LC-MS/MS. Based on this, pure artificial peptides were synthesized, and the efficacy of the selected peptides was verified using cellular and hyperlipidemic mouse models. Soybean hydrolysates, including two bioactive peptides (ALEPDHRVESEGGL and SLVNNDDRDSYRLQSGDAL), promoted TICE via the expression of ABCG5 and ABCG8 in enterocytes. They downregulated expression of hepatic cytochrome P450 family 7 subfamily A member 1 (CYP7A1) and CYP8B1 via expression of fibroblast growth factor 19 (FGF19) in a liver X receptor α (LXRa)-dependent pathway. Administration of bioactive peptides to hyperlipidemic mouse models by oral gavage reduced cholesterol levels in serum via upregulation of ABCG5 and ABCG8 expression in the proximal intestine and through fecal cholesterol excretion, upregulated FGF 15/19 expression, and suppressed hepatic bile acid synthesis. Oral administration of soybean-derived bioactive peptides elicited hypolipidemic effects by increasing TICE and decreasing hepatic cholesterol synthesis.

## 1. Introduction

Hyperlipidemia is characterized by the elevated circulation of very low-density and low-density lipoprotein cholesterol (VLDL-C and LDL-C) and decreased circulation of high-density lipoprotein cholesterol (HDL-C) in the blood. It is closely correlated with obesity and a sequence of cardiometabolic syndrome which includes hypertension and heart disease [1]. In the case of atherosclerosis, inordinate circulation of LDL-C is associated with atherosclerotic lesions, whereas decreased circulation of LDL-C delays the development of atherosclerosis [2]. Under other conditions, HDL particles play an essential role in the anti-atherosclerotic effect through acceptance of cholesterol and its transfer to the liver. HDL has an anti-oxidative role through the oxidation of LDL particles, and prevents the formation of atheroma in the sub-endothelial region [3]. Although HDL has a prophylactic role in LDL progression, it plays a minor role in the progression of hyperlipidemia. Therefore, hyperlipidemia has become an urgent health issue.

There are therapeutic strategies for hyperlipidemia. For instance, statins, and inhibitors of 3-hydroxy-3-methylglutaryl-coenzyme A (HMG-CoA) are commonly used to inhibit cholesterol synthesis and to decrease triglyceride (TG) and cholesterol levels in the blood. On the other hand, omega-3-fatty acids, fibrates, and niacin are commonly used as treatment options in statin-tolerant patients [4]. In previous studies, HMG-CoA has been shown to be an important enzyme in the cholesterol-related pathway, and its enzymatic products, including mevalonate, have shown physiological roles in other pathways [5]. Additionally, inappropriate statin prescriptions can result in diabetes mellitus, central nervous system disorders, and statin-associated muscle symptoms [6]. To ameliorate these adverse effects of statin therapies, combination therapy with ezetimibe is widely used and has shown enhanced LDL-C-lowering effects and improvement of LDL-C levels [7]. In addition, proprotein convertase subtilisin/kexin type 9 (PCSK9) inhibitors (evolocumab and alirocumab), benzoic acid, and a combination of bempedoic acid and ezetimibe, evinacumab, and other TG-lowering agents (e.g., icosapent ethyl) have emerged [8]. Although therapeutic strategies involving statin and non-statin therapies have improved, they are still insufficient for ameliorating the effects of hyperlipidemia.

The liver is a critical organ for cholesterol synthesis and excretion to the intestinal lumen; however, around 95% of cholesterol excretion via hepatobiliary cholesterol excretion is absorbed by the intestine [9]. Previous studies have shown that routes for cholesterol secretion via hepatobiliary transport and trans-intestinal cholesterol secretion or excretion (TICE), which are direct transport pathways through intestinal enterocytes [10]. The previous studies show that TICE-mediated cholesterol transport accounts for approximately 40% of cholesterol excretion to feces; thus, TICE is a suitable therapeutic target for cardiovascular diseases [11,12]. Based on molecular mechanisms, cholesterol of lipoprotein particles is accepted at basolateral enterocytes. Next, the ATP-binding cassette transporter G5 (ABCG5) and ABCG8 (heterodimers) facilitate cholesterol excretion into the intestinal lumen [13]. Because TICE could be a therapeutic target for hyperlipidemia, more effective and less adverse regulators of TICE are needed for the treatment of hyperlipidemia [14,15].

In the digestive process, proteins are digested through peptic and tryptic hydrolysis in the stomach and small intestine. The digested proteins yield individual amino acids. These protein hydrolysates have various bioactivities. The bioactivity of protein hydrolysates was investigated via analysis of their sequences. In addition, the bioactivity showed longevity effects despite ingestion of polypeptides [16]. Bioactive polypeptides have diverse functions, including anti-cancer [17], hypertensive [18], and immunoregulatory effects [19]. In addition, our previous study showed that casein-derived bioactive peptides affect TICE and bile acid metabolism [20].

Soy is a representative functional food, and its hydrolysate has been reported to be able to affect lipolysis in adipocytes [21] and the gut microbiome [22], and to have antihypertensive effects [23]. However, there are only a few studies on the bioactive peptides of soy hydrolysate and the mechanisms underlying their effect on hyperlipidemia. In the present study, we investigated the biological function and mechanisms of soy hydrolysates. Peptides from soy hydrolysates affect blood cholesterol levels by regulating TICE and bile acid metabolism, as observed in cellular and mouse models. Therefore, we elucidated that bioactive peptides from soy hydrolysates have a promising therapeutic role in hyperlipidemia.

## 2. Materials and Methods

### 2.1. Chemicals, Antibodies, and Reagents

Soybean powder, trypsin, and pepsin for soy hydrolysis were purchased from Sigma Aldrich (St. Louis, MO, USA). Monoolein and sodium taurocholate for TICE assay were purchased from Sigma Aldrich (St. Louis, MO, USA). siRNA for control and human FGF19 were purchased from Bioneer (Daejeon, Korea). Antibodies specific for ABCG5 and ABCG8 were purchased from Abcam (Cambridge, MA, USA). FGF15, FGF19, GAPDH, and alpha-tubulin were purchased from Santa Cruz Biotechnology (Santa Cruz, CA, USA). Dulbecco’s modified Eagle’s medium (DMEM), Eagle’s minimum essential medium (MEM), fetal bovine serum (FBS), streptomycin, penicillin, and TRIzol were obtained from Thermo Fisher Scientific (Cleveland, OH, USA).

### 2.2. Cell Culture and Treatment

As previously described, the human colorectal cancer cell line Caco-2 and the human normal hepatocyte cell line MIHA were cultured [24]. Briefly, MEM (for Caco-2) and DMEM (for MIHA) were utilized supplemented with 10% FBS and penicillin (100 U/mL), and streptomycin (100 mg/mL), respectively. The cell incubator setting was 37 °C, with 5% CO_2_ and humidity. Before treatment, the cells were incubated in serum-free media for 24 h [25].

### 2.3. Soy Hydrolysis

For soybean hydrolysis, pepsin and trypsin treatments were performed as previously described [20]. Briefly, the soy solution was prepared at 5 mg/mL in distilled water. The pH of the soy solution was adjusted to approximately 2 by adding a 40% HCl solution and incubated with pepsin (0.4% weight per volume) at 37 °C for 2 h. Next, the pH of the solution was adjusted to 7.6 by adding a NaOH solution and incubated with trypsin (0.4% weight per volume) at 37 °C for 2 h. The hydrolysates were added with SDS buffer, loaded with sodium dodecyl sulphate–polyacrylamide gel electrophoresis (SDS-PAGE), and stained with Coomassie Blue.

### 2.4. Total RNA Isolation and qRT-PCR

For mRNA expression assessment, qRT-PCR was performed as described in a previous study [26]. Briefly, RNA was isolated using TRIzol, following the manufacturer’s instructions, and qRT-PCR was performed using an Applied Biosystems StepOne Real-Time PCR System (Applied Biosystems, Foster City, CA, USA) for 40 cycles at 95 °C for 15 s and at 60 °C for 1 min, followed by thermal denaturation. The primer sequences used are listed below (Table 1). Each sample was assessed in triplicate.

### 2.5. Western Blotting

For protein expression assessment, western blotting was utilized as described previously [27]. Briefly, whole cell lysates were prepared using radioimmunoprecipitation assay lysis buffer (50 mM Tris (pH 7.4), 1% Triton X-100, 150 mM NaCl, 1 mM dithiothreitol, 25 mM NaF, and 20 mM EGTA supplemented with protease inhibitors), and a Bio-Rad protein assay kit (Bio-Rad Laboratories, Hercules, CA, USA) was used to determine protein concentrations. Protein samples were subjected to SDS-PAGE, transferred to an NC (nitrocellulose) membrane, and then blocked with 5% BSA (bovine serum albumin) in TBST (100 mM NaCl, 10 mM Tris, and 0.1% Tween 20). The membranes were probed with specific primary antibodies overnight at 4 °C. Next, the membranes were washed in TBST and probed with peroxidase-conjugated secondary antibodies (Santa Cruz Biotechnology, Santa Cruz, CA, USA). The membranes were analyzed using an ECL detection system (Roche Applied Science, Indianapolis, IN, USA) with iBright chemi-doc fl000 from Thermo Fisher Scientific. The images of western blot data were quantified using ImageJ and validated by statistical analyses.

### 2.6. Cholesterol Assay

To measure the total cholesterol levels in cells, media, serum, and feces, a total cholesterol assay kit (Cell Biolabs, San Diego, CA, USA) was used. Following the manufacturer’s instructions, cells and feces were homogenized in an extraction solution with a mixture of chloroform: isopropanol:NP-40 of 7:11:0.1, centrifuged at 15,000× *g* for 10 min, and the supernatant was obtained. The solution was dried at 50 °C, and the dried lipids were dissolved in assay buffer. The media and serum were diluted in the assay buffer. The samples were then subjected to cholesterol assay and detected at 560 nm using a GloMax fluorescence detection system. Each sample was measured in triplicate.

### 2.7. In Vitro TICE Assay

Following a previous study, Caco-2 cells were incubated on the insert of the transwell and differentiated for 7 days [20,28]. To prepare a media containing cholesterol, MEM media was supplemented with monoolein (30 μM), sodium taurocholate (500 μM) and/or cholesterol (100 μM) and subsequently sonicated for 15 min to form micelles. To assess the in vitro TICE, the upper chamber was filled with media without cholesterol, and the lower chamber was filled with media containing cholesterol. The media in the upper chamber were harvested 24 h after peptide and GSK2033 treatment and applied to the cholesterol assay.

### 2.8. High-Performance Liquid Chromatography (HPLC) Analysis of Soy Hydrolysates

HPLC was used to separate peptides contained in the protein hydrolysates. A Waters 1525 Binary HPLC pump (Wasters, Milford, MA, USA), Sunfire C18 column (4.6 × 250 mm), and Waters 2489 UV/Visible detector (Waters) were used. The mobile phase was an isocratic combination of acetonitrile:H_2_O (50:50) at a 1 mL/min flow rate. The eluates were collected following the real-time UV detection results (214 nm).

### 2.9. Peptide Sequencing and Synthesis

To analyze the bioactive peptides contained in the HPLC eluates of soy hydrolysates, the bioactive fraction was applied to peptide identification liquid Chromatography with tandem mass spectrometry (LC-MS/MS) performed by Life Science Laboratory. Co. (http://www.emass.co.kr, 25 June 2021), Seoul, Korea. Depending on the peptide identification results, artificial peptides were synthesized and prepared by Peptron Co. (http://peptron.co.kr, 25 June 2021), Daejeon, Korea.

### 2.10. Cellular Viability Assay

To measure the cellular toxicity of peptides, CellTiter-Glo^®^ Luminescent Cell Viability Assay kit (Promega, Madison, WI, USA) was used. Following the manufacturer’s instructions, cells were seeded and incubated in a 96-well plate. Cells were treated with the bioactive peptides 24 h prior to detection. The samples were detected using a GloMax luminescence detection system. Each sample was measured in triplicate.

### 2.11. Animal Care Protocol

Six-week-old male C57BL/6 mice (Orient Bio, Seongnam, Korea) were used for the in vivo experiments, based on protocols specified in a previous study [29]. The protocols used were approved by the Institutional Animal Care and Use Committee of Pusan National University (Busan, Korea) and performed in accordance following the National Institutes of Health Guide for the Care and Use of Laboratory Animals. The mice were housed individually or in groups of up to five mice in sterile cages. They were maintained in animal care facilities at room temperature (23 °C ± 1 °C) with a 12-h light-dark cycle. The animals were fed water and a standard mouse chow diet or a high cholesterol diet (HCD) ad libitum. The animal protocol used in this study was approved by the Pusan National University Institutional Animal Care and Use Committee (PNU-IACUC) for ethical procedures and scientific care (Approval Number PNU-2020-2809) on 2 December 2020.

Before the experiment, the mice were randomly divided into experimental groups (*n* = 10). To establish hyperlipidemia and assess peptide effects, the mice were fed with HCD (21% milkfat, 0.5% cholic acid, and 1.25% cholesterol), and peptides were orally administered at 200 μg/day for 7 weeks. At the end of the administration, the mice were anesthetized with isoflurane for inhalational anesthesia and perfused. The blood, liver, and small intestine (divided into three parts: the proximal part of the small intestine, which attaches to stomach; the middle, between the proximal and distal parts; and the distal, the part of the small intestine which attaches to colon) as well as the feces were harvested.

### 2.12. Enzyme-Linked Immunosorbent Assay (ELISA)

To assess secretory FGF15/19 levels in serum and media samples, an indirect ELISA was performed. The samples were attached to 96-well immunoplates (SPL, Seoul, Korea), blocked with 1% BSA in PBS, and probed with primary antibodies and HRP-conjugated secondary antibodies. TMB (3,3’,5,5’-Tetramethylbenzidine) was utilized and detected at 450 nm. Each sample was assessed in triplicate.

### 2.13. Statistical Analysis

All numerical data are presented as the mean ± standard error from at least three independent experiments. For quantification, data were analyzed using *t*-test, One-way ANOVA and multiple comparison (Dunnett’s T3 test and Tukey test). Prism 9 software (GraphPad Software, San Diego, CA, USA) was used for all statistical analyses. Statistical significance was set at *p* < 0.05.

## 3. Results

### 3.1. Soy Hydrolysates Upregulate TICE and Downregulate Cholesterol Levels

As shown in a previous study, hydrolysis through the digestive system contributes to the bioactivity of soybean [30]. To elucidate the effects of hydrolyzed soybean, we produced soy hydrolysates using highly purified isolated soybean powder (minimum protein content of 90%) in distilled water. Then, we incubated soy solution with pepsin and trypsin at body temperature and a pH range of pH 2–3 and pH 7–8, respectively. After incubation, the digested solution was validated using SDS-PAGE and Coomassie blue staining (Figure 1A). There was no detection for negative control (soy solution), but small peptides from soy hydrolysate were detected. To confirm that soy hydrolysates regulate TICE, we utilized an in vitro small intestine model through the Caco-2 cell line as previously described [28]. Soy protein or hydrolysates were applied to Caco-2 cells at 2 mg/mL for 24 h, and we assessed *ABCG5* and *ABCG8* mRNA expression [31]. Soy protein and soy hydrolysate upregulated *ABCG5* and *ABCG8* mRNA levels (Figure 1B). Additionally, ABCG5/8 protein levels are increased via soy solution treatment (not hydrolysis), and soy hydrolysate increased their expression to a greater degree than soy treatment (Figure 1C). Next, we assessed the effect of soy hydrolysates on cholesterol regulation. Soy protein and soy hydrolysate increased the TICE amount via topical cholesterol transport by approximately 30% and 80%, respectively (Figure 1D). Next, to elucidate the effect of soy hydrolysate in vivo, we used a high-cholesterol diet (HCD) to generate a hyperlipidemic mouse model. We orally administrated soy hydrolysates to the HCD mice for three weeks. As a result, administration of soy hydrolysate decreased serum cholesterol level by approximately 15% compared with mice fed only on an HCD (Figure 1E). Consequently, the results show that the digestive product of soybean induced cholesterol excretion in vitro and decreased cholesterol levels in serum.

### 3.2. Soy Hydrolysate-Derived Bioactive Peptides Induce TICE

In previous studies, ingestion of bioactive peptides was noted to have biological effects [32]. Although the mechanism of bioactive peptides for effects on biological processes have not been clarified, a previous study on bioactive peptide has suggested that amino acid sequences are important for the effects of these peptides [33]. Based on the importance of amino acid sequences, we hypothesized that soy and soy hydrolysates exert hypolipidemic effects through specific bioactive peptides arising from soybean digestion. Using HPLC, 2 mg of soy hydrolysates was divided into three fractions based on their hydrophobicity (water:acetonitrile ratio, Figure 2A). To elucidate the important fraction for the hypolipidemic effects, we treated the fraction and assessed the level of *ABCG5* and *ABCG8* in Caco-2 cells. We observed that only fraction #2 upregulated levels of ABCG5 and ABCG8 (Figure 2B,C). Next, we further analyzed fraction #2 using LC-MS/MS-based peptide identification. Consequently, we discovered 11 peptide sequences in fraction #2 (Table 2). To prove the effects of these 11 synthetic peptides for ABCG5 and ABCG8 regulation, we performed analysis by using 1 μg/mL in distilled water of each peptide to treat Caco-2 cells [20,33]. We confirmed that peptides 1 and 8 significantly upregulated ABCG5 and ABCG8 expression by 1.5-fold (Figure 2D). We further examined cell viability via the peptide treatment utilizing cellular luminescence assay. As a result, treatment of the peptides could not impair cell viability in Caco-2 cells (Figure 2E). These results show that soy hydrolysates have bioactivity and exert hypolipidemic effects through specific bioactive peptides.

### 3.3. Soybean-Derived Peptides Upregulate TICE via LXRα Signaling

To elucidate how TICE is regulated by peptides, we confirmed the TICE amount in vitro through the treatment of each peptide. Consequently, peptide 1 and peptide 8 induced in vitro TICEs (Figure 3A). In addition, peptides 1 and 8 upregulated ABCG5 and ABCG8 protein levels (Figure 3B). Next, to validate the signaling pathways induced via the peptide treatment to increase intestinal ABCG5 and ABCG8 levels, we analyzed the liver X receptor α (LXRα) signaling. As previous studies have reported, ABCG5 and ABCG8 are transcriptional targets of LXRα [34]. As LXRα is the primary inducer of ABCG5 and ABCG8, we utilized GSK2033, a specific LXRα antagonist, to modulate ABCG5 and ABCG8 expression. Treatment of Caco-2 cells with GSK2033 resulted in a 30% reduction in ABCG5 and ABCG8 levels and was not rescued by peptide treatment (Figure 3C). Additionally, GSK2033 treatment downregulated ABCG5 and ABCG8 protein expression (Figure 3D). Similarly, GSK2033 treatment reduced the in vitro TICE amount and decreased the effects of peptide treatment (Figure 3E). These results show that soybean-derived peptides 1 and 8 upregulate cholesterol excretion via LXRα-mediated ABCG5 and ABCG8 levels.

### 3.4. Bioactive Peptides Regulate Bile Acid Synthesis via Regulation of Enterocyte-Derived FGF19

In fecal cholesterol excretion, TICE has a one-third proportion; hepatobiliary cholesterol transport is also critical for cholesterol excretion to feces and hypolipidemic strategy [35]. As previously described, in vivo TICE regulated intestinal bile acid profiles modulated via metabolic change of hepatic bile acid [12]. Intestine-derived secretary factors regulate bile acid metabolism in the liver. In addition, secretary factors are important for the regulation cycle of bile acid in the liver and intestine. Fibroblast growth factor 19 (FGF19) is a typical intestine-derived secretory protein and has modulating effects on the metabolic pathway of bile acid in the liver. Therefore, we assessed whether FGF19 expression is altered by peptide treatment and farnesoid X receptor (FXR) level. In previous studies, FXR was found to play a role in FGF19 expression and TICE [12]. We observed that FGF19 expression was upregulated by peptide treatment, while FXR expression remained unchanged (Figure 4A). Increased FGF19 secretion was observed in the culture medium (Figure 4B). We confirmed that the LXRα signaling pathway is mediated by peptides 1 and 8 (Figure 3) and that the LXRα ligand increased the expression of intestinal FGF19 [36]. Therefore, we assessed whether GSK2033 and peptide treatment could alter *FGF19* and *FXR* expression. As a result, the expression of *FGF19* was significantly downregulated, while GSK2033 treatment barely rescued that of FXR with or without peptide treatment (Figure 4C). In addition, we confirmed that GSK2033 suppressed the secretion of FGF19 and that peptide treatment could not rescue the secretion (Figure 4D). To validate regulation of FGF19 via peptide for the metabolic pathway of bile acid in liver, conditioned media (CM) from the peptide-treated Caco-2 cells was added to MIHA cells. We confirmed the level of cytochrome P450 family 7 subfamily A member 1 (*CYP7A1*) and *CYP8B1*, which are major cholesterol synthesis-related genes. The expression of CYP7A1 and CYP8B1 was reported to be downregulated by ileal FGF19 secretion [12]. We observed that CM suppressed *CYP7A1* and *CYP8B1* expression (Figure 4F). To validate the effect of FGF19 on *CYP7A1* and *CYP8B1* levels in the liver, CM from *FGF19* siRNA-treated was added to Caco-2 cells (Figure 4E). We showed that it rescued the downregulation of *CYP7A1* and *CYP8B1* levels (Figure 4F). Moreover, peptides obtained via soybean digestion modulated the hepatic bile acid synthetic pathway via FGF19 secretion.

### 3.5. Bioactive Peptides Attenuate Cholesterol-Derived Obesity and Hyperlipidemia

Owing to our observations of the effects of bioactive peptides on TICE and hepatic bile acid metabolism in vivo, we established hyperlipidemic mouse models using an HCD. The mice were administered peptide 1 or 8 five times orally at 10 mg/kg in a week [37]. To investigate the prophylactic and therapeutic effects of soybean-derived peptides, peptides were orally injected with HCD. Peptide treatment diminished the weight of mice by approximately 25% after seven weeks of administration (Figure 5A). To confirm the hypolipidemic effects of peptide treatment, we confirmed the cholesterol levels in serum and feces. We observed that peptide treatment decreased serum cholesterol levels by approximately 33% and increased fecal cholesterol levels by approximately 50% after seven weeks of administration (Figure 5B). According to a previous study, TICE occurs in the proximal intestine [10]. Therefore, we confirmed *Abcg5*/*8* levels in proximal and distal intestines to validate the effect of peptide administration in the intestine. In the proximal intestine, peptide treatment increased *Abcg5* and *Abcg8* expression (Figure 5C). However, levels of *Abcg5* and *Abcg8* were unaltered via peptide treatment in the distal intestine (Figure 5C). We quantified Abcg5/8 protein levels using western blotting. Peptide treatment upregulated Abcg5/8 protein levels (Figure 5C). We previously assessed the intestinal expression of *FGF19* and *FXR* in vitro (Figure 4A). Next, we confirmed the level of *Fgf15* (*FGF19* murine homolog form) and *Fxr*. We observed that peptide administration did not alter the level of *Fxr* in the proximal intestine. The level of *Fgf15* was significantly increased by the peptide treatment in HCD mice (Figure 5D). These results are consistent with our previous in vitro results. Next, we found that serum *Fgf15* levels were downregulated by 20% in the HCD group and rescued by the peptide treatment (Figure 5E). The downregulation of serum *Fgf15* levels demonstrated that *Fgf15* might have a role in the increase of systemic *Fgf15* circulation. Finally, to confirm whether increased Fgf15 expression plays a role in the metabolic pathway of bile acid, we assessed levels of CYP7A1 and CYP8B1 in the liver. The HCD group showed reduced *Cyp7a1* and *Cyp8b1* levels (Figure 5F). The peptide treatment further diminished these changes. Collectively, soybean-derived bioactive peptides 1 and 8 had weight-reducing effects and hypolipidemic effects in the in vivo model. Specifically, bioactive peptides upregulated the Abcg5/8 level in the proximal intestine, thereby upregulating cholesterol excretion by TICE. In addition, peptides 1 and 8 upregulated Fgf15 secretion, further decreasing cholesterol synthesis correlated with Cyp7a1 and Cyp8b1 levels (Figure 6).

## 4. Discussion

As the severity of hyperlipidemia and its complications are both increasing, therapeutic strategies for hypolipidemia must be developed. In addition to previous studies and other therapeutic strategies, the promotion of TICE may increase treatment efficacy [12]. In this study, we demonstrated that two specific soybean-derived peptides (peptide 1, ALEPDHRVESEGGL, and peptide 8, SLVNNDDRDSYRLQSGDAL) could upregulate TICE by inducing ABCG5 and ABCG8 expression and LXRα signaling activation. In addition, we confirmed that secretion of FGF15/19 from enterocytes was increased via peptides 1 and 8, which reduced hepatic bile acid synthesis to support hepatobiliary cholesterol excretion. These results indicate that peptides formed during the digestive process have bioactivity associated with the regulation of systemic cholesterol homeostasis.

In the context of cholesterol regulating strategies, TICE has been studied as an adjuvant cholesterol-lowering pathway for hepatobiliary cholesterol excretion. Given that TICE was noted to induce approximately one-third of cholesterol excretion, it has been considered to have clinical potential for hyperlipidemia treatment [35]. Our study showed that peptides from dietary soybean can upregulate TICE by increasing ABCG5 and ABCG8 expression. Based on the results of treatment with GSK2033, a specific LXRα antagonist, it can be concluded that the transcriptional activity of LXRα mediates peptide-induced ABCG5 and ABCG8 expression. In a previous study, LXRα was associated with ABCG5 and ABCG8 expression and induction of TICE, and we observed that the induction of signaling pathways by soybean-derived peptides was involved in LXRα activity [38]. In addition, we observed that peptide treatment upregulated ABCG5 and ABCG8 expression in only the proximal intestine. Consistent with previous studies, TICE steadily decreased upon movement toward the distal intestine and peptide-derived TICE increased only in the proximal intestine. Therefore, peptide-mediated ABCG5 and ABCG8 upregulation effectively increased fecal cholesterol excretion [10,20]. Our in vivo results showed that peptide 1 and 8 downregulated the serum cholesterol levels while increasing the fecal cholesterol levels. To clarify whether the hypolipidemic effect of the peptides is caused by ABCG5/8-mediated TICE, further study is needed to show that cholesterol levels remain unchanged by peptide treatment in ABCG5/8 knock-out mice. Our results showed that the bioactive peptides generated upon soybean digestion increase TICE in an LXRα-dependent manner.

Proteins are divided into amino acids via digestion processes, including digestive enzymatic functions. Moreover, the amino acids are absorbed in the small intestine and affect biological processes [16,17,18,19,20]. In this current study, we discovered two bioactive peptides, peptides 1 and 8, which are approximately 1.5 kDa and 2.1 kDa in size, respectively. Peptides 1 and 8 have not been reported to date. The original protein of peptide 1 is glycinin, while peptide 8 is a beta-conglycinin alpha subunit. Glycinin and beta-conglycinin alpha subunits are known storage proteins [39]. Although glycinin and beta-conglycinin are allergenic proteins in humans, only their acidic and macro polypeptides are known to induce allergenic symptoms [40,41]. In a previous study, soybean glycinin improved HDL-C level and atherogenic index when used in a hypercholesterolemic chow diet [42]. Similarly, a soybean beta-conglycinin diet suppressed serum TG levels by decreasing fatty acid synthase expression and suppressing TG absorption and beta-oxidation in mice [43]. As shown in previous studies, the effects of soybean-derived glycinin and beta-conglycinin on the attenuation of lipid levels need to be investigated with respect to the underlying molecular mechanisms. In addition, further understanding of bioactive peptide characteristics is needed in order to evaluate the effects of other biological processes. Therefore, the current study provides a reasonable framework for understanding hyperlipidemic symptoms.

In our in vitro and in vivo experiments, treatment with peptides 1 and 8 induced inhibition of CYP7A1 and CYP8B1 hepatic expression by upregulating FGF15/19 levels and secretion. In the bile acid synthesis, CYP7A1 is a rate-limiting enzyme and CYP8B1 has an important role in the homeostasis of cholic acid (CA) and chenodeoxycholic acid (CDCA) in the liver; moreover, CYP7A1 and CYP8B1 regulate the levels of synthetic cholesterol [44,45]. This study elucidated that hepatic expression of CYP7A1 and CYP8B1 is downregulated in hyperlipidemic mouse models and that suppression of FGF15/19 induces a decrease in CYP7A1 and CYP8B1. Recently, it was reported that the modulation of the FGF15/19 pathway affects proliferation and metabolic function in hepatocytes, intestinal FGF15/19 physiologically inhibits hepatic lipogenesis, and FGF15/19 controls hepatic cholesterol and bile acid homeostasis [46,47,48]. Furthermore, regulating FGF15/19 affects carbohydrate and lipid metabolism, including TG concentrations, insulin sensitivity, weight loss, and obesity-associated hyperlipidemia [49]. The results of these studies are consistent with those of the current study, given that Fgf15 expression changed in hyperlipidemic mouse models. In conclusion, soybean-derived peptides 1 and 8, via modulation of FGF15/19 expression, induce TICE and regulate systemic lipid metabolism. Collectively, these results suggest that peptides 1 and 8 are potential therapeutic targets for obesity and hyperlipidemia.

## 5. Conclusions

We discovered two efficient bioactive peptides from soybean and illuminated the mechanisms involved in hypolipidemic effects. As soybean is a widely consumed food, the bioactivities of peptides generated by its digestion were analyzed using artificial synthetic peptides; moreover, soybean-derived peptide sequences can be used in further studies to enhance the effectiveness of peptides and investigate other cholesterol-related molecular mechanisms. Lastly, further exploration of safe food ingredients in biological processes can help identify alternative therapeutic strategies to prevent adverse effects.

## Figures and Tables

**Figure 1 nutrients-14-00095-f001:**
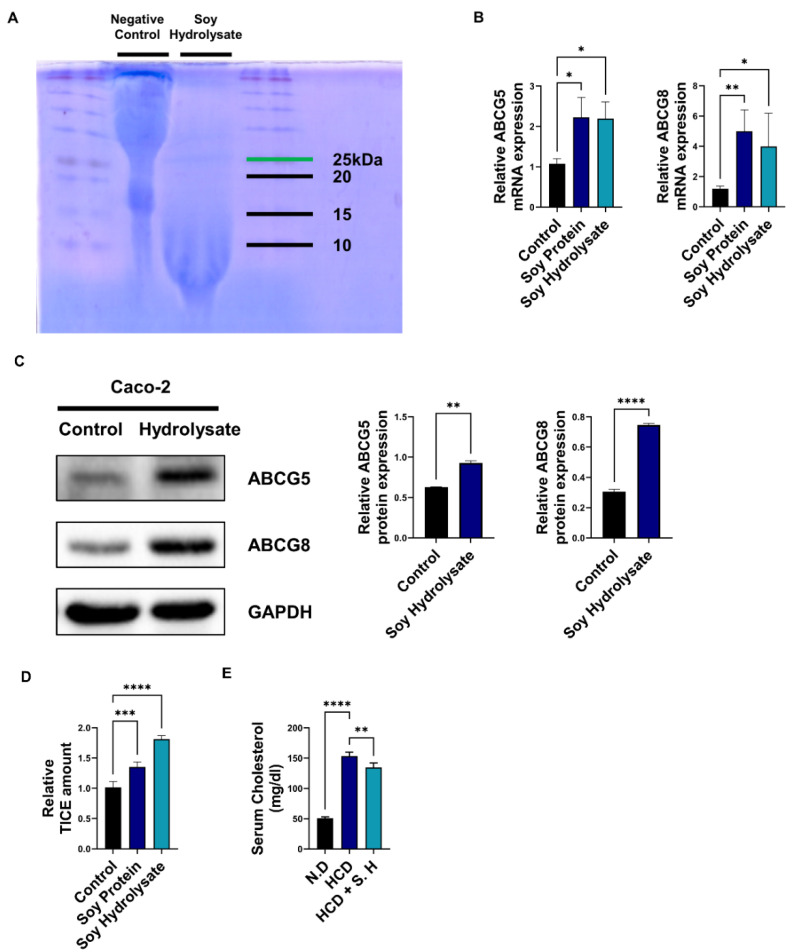
Soy hydrolysates attenuate hyperlipidemia and induce TICE. (**A**) Coomassie blue staining of the soy protein hydrolysis by digestive enzymes pepsin and trypsin. (**B**,**C**) The mRNA and protein level of ABCG5/8 in soy protein or soy hydrolysates (2 mg/mL) treated Caco-2 cells. (**D**) The relative TICE amount in soy protein or soy hydrolysate treated Caco-2 cell via cholesterol assay. (**E**) Using cholesterol assay, serum cholesterol levels in mice feeding a high-cholesterol diet (HCD) or high-cholesterol diet + soy hydrolysate (HCD + S. H) (5 mg/day). *, *p* < 0.05. **, *p* < 0.01. ***, *p* < 0.001. ****, *p* < 0.0001. ABCG5/8, ATP-binding cassette subfamily G member 5/8; GAPDH, glyceraldehyde 3-phosphate dehydrogenase; N.D., normal diet; HCD, high-cholesterol diet; S.H., soy hydrolysates.

**Figure 2 nutrients-14-00095-f002:**
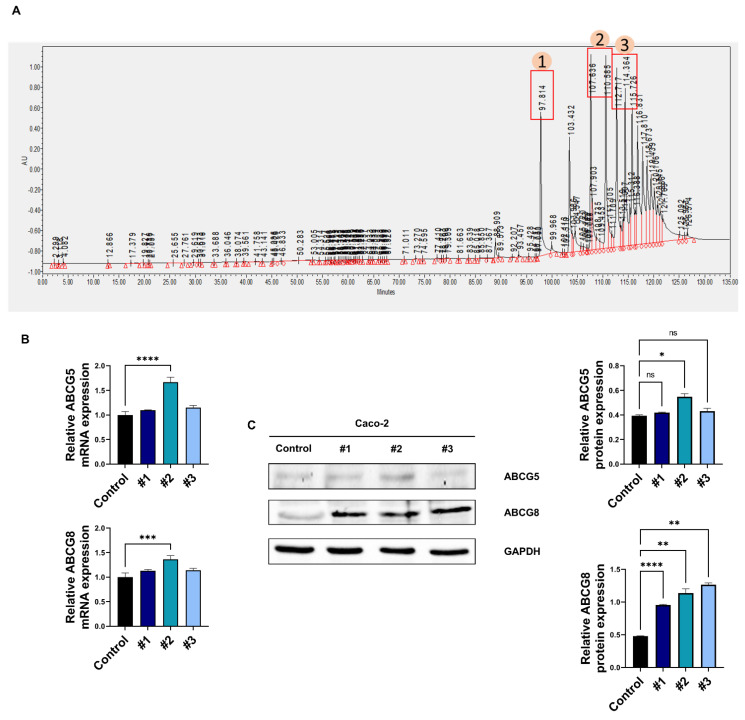
Bioactive peptides in soy hydrolysates increase ABCG5 and ABCG8. (**A**) The results of HPLC analysis. Utilizing acetonitrile as the flowing solvent, the percentage of acetonitrile progressively increased to 50% for around 2 h. The eluates of HPLC were divided into three fractions and numbered #1, #2, and #3 in order. (**B**,**C**) The mRNA and protein level of ABCG5/8 in HPLC eluate-treated Caco-2 cells. (**D**) The mRNA level of *ABCG5/8* in 11 synthetic peptides applied to Caco-2 cells. (**E**) Using CellTiter-Glo^®^ Luminescent Cell Viability Assay, the relative cellular viability in peptide 1 or 8-treated Caco-2 cells. *, *p* < 0.05. **, *p* < 0.01. ***, *p* < 0.001. ****, *p* < 0.0001. HPLC, high performance liquid chromatography. ns, no significant.

**Figure 3 nutrients-14-00095-f003:**
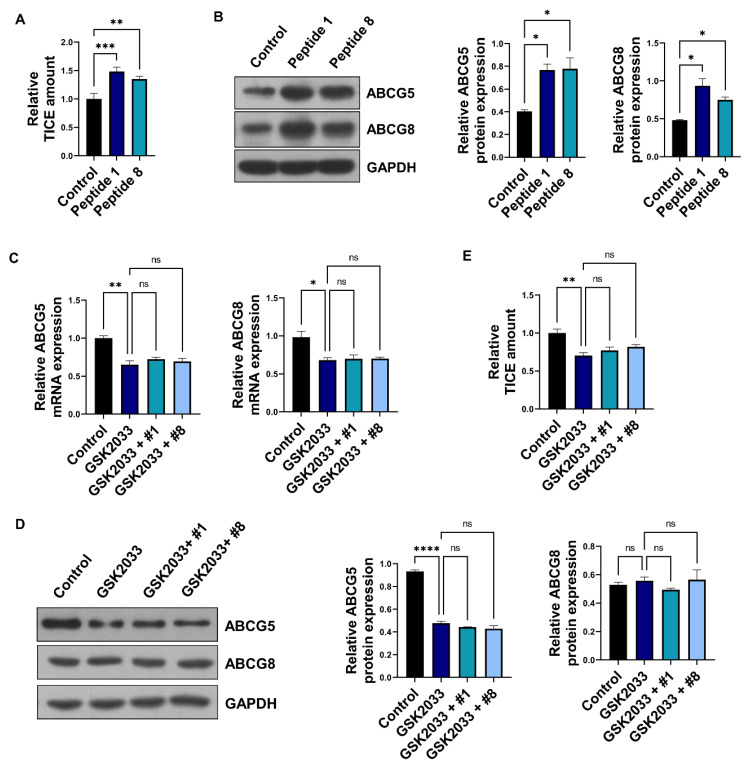
Soybean-derived peptide upregulates TICE via LXRα-dependent manner. (**A**) The relative TICE amount in peptide 1 or 8-treated Caco-2 cells via cholesterol assay. (**B**) The protein expression of ABCG5/8 in peptide 1 or 8-treated Caco-2 cells. (**C**,**D**) The mRNA and protein expression of ABCG5/8 in GSK2033 (1 μM) and peptide 1 or 8-treated Caco-2 cells. (**E**) Using cholesterol assay, the relative TICE amount in GSK2033 (1 μM) and peptide 1 or 8-treated Caco-2 cells. *, *p* < 0.05. **, *p* < 0.01. ***, *p* < 0.001. ****, *p* < 0.0001. GSK, LXR antagonist. ns, no significant.

**Figure 4 nutrients-14-00095-f004:**
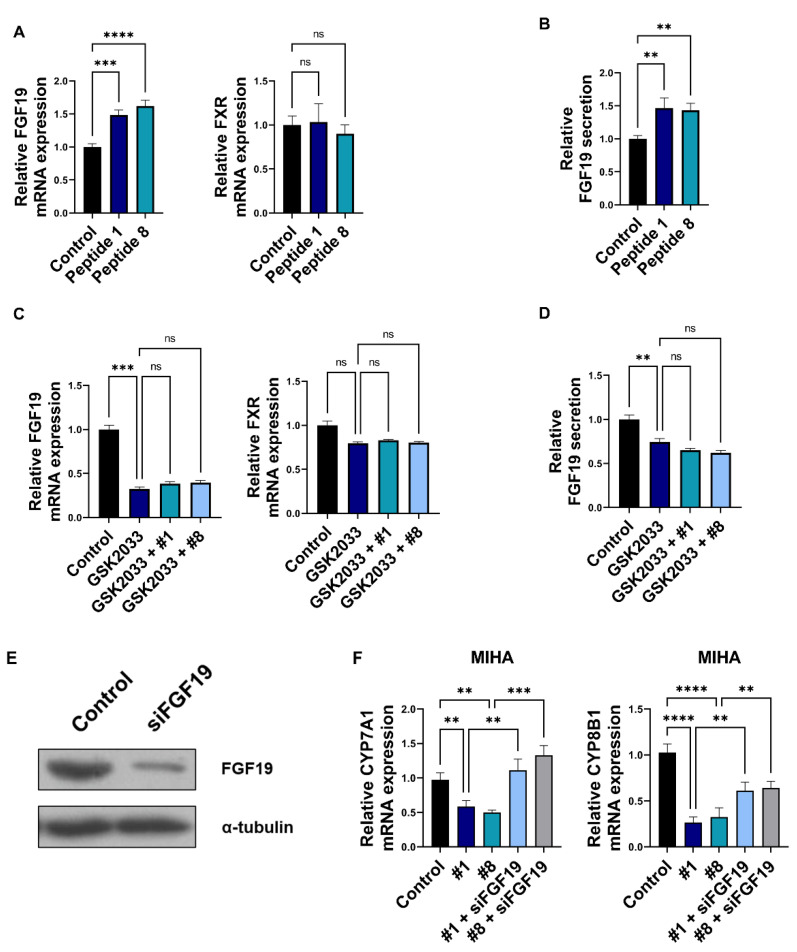
FGF19 from enterocytes changes bile acid metabolism in the small intestinal lumen. (**A**) The mRNA level of *FGF19* and *FXR* in peptide 1 or 8-treated Caco-2 cells. (**B**) Using ELISA, the changes of secretory FGF19 level in conditioned media of peptide 1 or 8-treated Caco-2 cells. (**C**) The mRNA expression changes of *FGF19* and *FXR* in GSK2033 and peptide 1 or 8-treated Caco-2 cells. (**D**) Using ELISA, the alteration of secretory FGF19 level in conditioned media of GSK2033 and peptide 1 or 8-treated Caco-2 cells. (**E**) Immunoblotting analysis of FGF19 expression in Caco-2 cells with control or FGF19 siRNAs. (**F**) The changes of *CYP7A1* and *CYP8B1* in conditioned media (treatment of peptide 1 or 8 and FGF19 siRNA Caco-2)-treated MIHA cells. **, *p* < 0.01. ***, *p* < 0.001. ****, *p* < 0.0001. MIHA, human hepatocytes cell lines; FGF, fibroblast growth factor. ns, no significant.

**Figure 5 nutrients-14-00095-f005:**
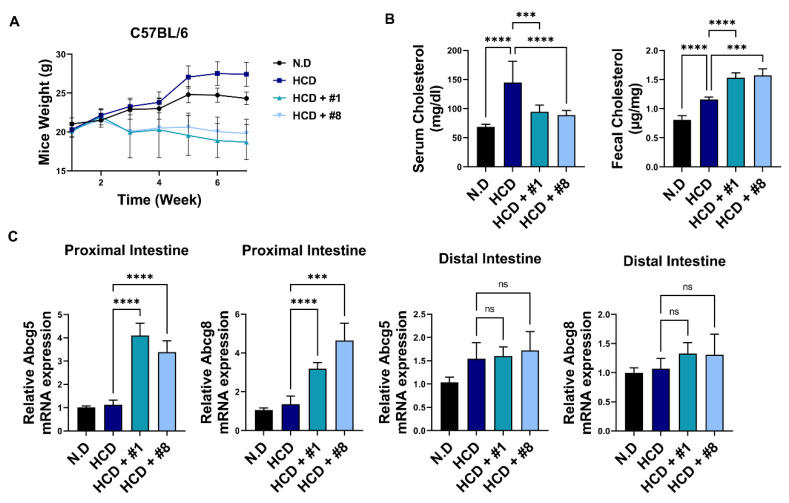
Soybean-derived peptide attenuates hyperlipidemia (**A**) The changes in the weight of mice from HCD diet and/or peptide administration in mice. (**B**) The cholesterol levels of HCD-diet mice and/or peptides in serum and feces. (**C**) The mRNA and protein expression of Abcg5/8 from HCD diet and/or peptide administration in the proximal or distal intestine. (**D**) The level of *Fgf15* and *Fxr* from HCD diet and/or peptide administration in the proximal or distal intestine. (**E**) Using ELISA, the serum Fgf15 level from HCD diet and/or peptide administration in serum. (**F**) The level of *Cyp7a1* and *Cyp8b1* from HCD diet and/or peptide administration in the liver. *, *p* < 0.05. **, *p* < 0.01. ***, *p* < 0.001. ****, *p* < 0.0001. HCD, high-cholesterol diet; N.D., normal diet. ns, no significant.

**Figure 6 nutrients-14-00095-f006:**
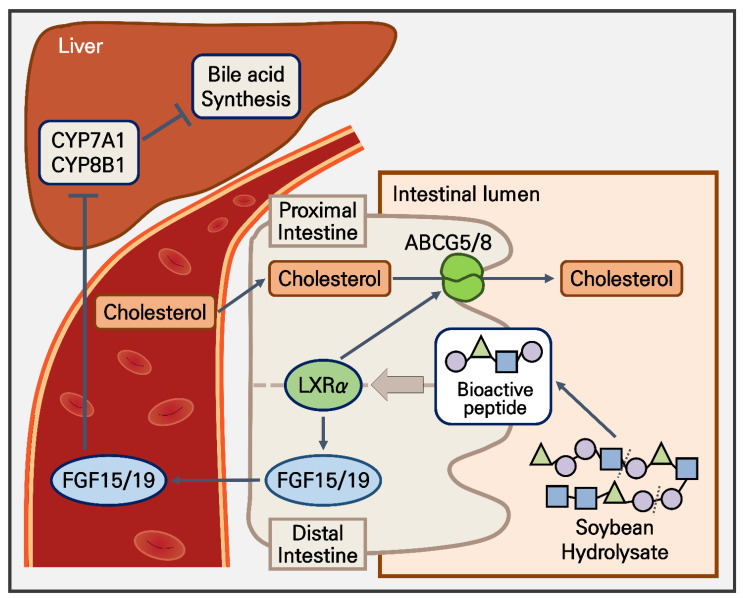
Graphical abstract of bioactive peptides from soybean effects on hyperlipidemia. The production of bioactive peptides via enzymatic hydrolysis and hypolipidemic effects. CYP7A1, cytochrome P450 family 7 subfamily A member 1; CYP8B1, cytochrome P450 family 8 subfamily B member 1; FGF, fibroblast growth factor.

**Table 1 nutrients-14-00095-t001:** Primers used for qRT-PCR.

Gene Name	Forward Primer	Reverse Primer
Human *ABCG5*	5′-AGCAAGGAACGGGAAATAGA-3′	5′-CAGGAGAACACCCAGTTTAGAG-3′
Human *ABCG8*	5′-GATACAGCCGCCCTCTTTT-3′	5′-GCCCGTCTTCCAGTTCATAG-3′
Human *FGF19*	5′-AGATCAAGGCAGTCGCTCTG-3′	5′-AAAGCACAGTCTTCCTCCGA-3′
Human *FXR*	5′-AAAGTTGTGTAAGATTCACCAGCCT-3′	5′-GGTCGTTTACTCTCCATGACATCA-3′
Human *CYP7A1*	5′-GACCACATCTTTGATTTGG-3′	5′-CCGTTTGCCTTCTCCTAA-3′
Human *CYP8B1*	5′-GCCTGTCCTTTGTAATGCTGA-3′	5′-GAAGCGAAAGAGGCTGTCC-3′
Human *GAPDH*	5′-ATGACATCAAGAAGGTGGTG-3′	5′-CATACCAGGAAATGAGCTTG-3′
Mouse *Abcg5*	5′-CTTCGACAAAATTGCCATCC-3′	5′-GAAAGGAACCGTGGGTAAGG-3′
Mouse *Abcg8*	5′-TGGTCAGTCCAACACTCTGG-3′	5′-ACTGGGTTGCCCATTTATCC-3′
Mouse *Fgf15*	5′-GAGGACCAAAACGAACGAAATT-3′	5′-ACGTCCTTGATGGCAATCG-3′
Mouse *Fxr*	5′-AAATGAGGGCTGCAAAGGTTTCT-3′	5′-TGCCCCCGTTCTTACACTTG-3′
Mouse *Cyp7a1*	5′-TACAGAGTGCTGGCCAAGAG-3′	5′-GCTGTCCGGATATTCAAGGA-3′
Mouse *Cyp8b1*	5′-CCTCTGGACAAGGGTTTTGTG-3′	5′-GCACCGTGAAGACATCCCC-3′
Mouse *Gapdh*	5′-CGACTTCAACAGCAACTCCCACTCTTCC-3′	5′-TGGGTGGTCCAGGGTTTCTTACTCCTT-3′

**Table 2 nutrients-14-00095-t002:** Peptide sequence contained in fraction #2.

No.	Sequence	Original Protein
1	ALEPDHRVESEGGL	Glycinin
2	NALEPDHRVESEGGL	Glycinin
3	FVDAQPQQKEEGN	Beta-conglycinin alpha’-subunit
4	VDAQPQQKEEGN	Beta-conglycinin alpha’-subunit
5	VVNPDNDENLRM	Beta-conglycinin alpha’-subunit
6	YVVNPDNDENLRM	Beta-conglycinin alpha’-subunit
7	SLVNNDDRDSY	Beta-conglycinin alpha-subunit
8	SLVNNDDRDSYRLQSGDAL	Beta-conglycinin alpha-subunit
9	VGLKEQQQEQQQEEQPLEVR	Beta-conglycinin alpha-subunit
10	TISSEDEPFNLRS	Beta-conglycinin beta-subunit
11	FPFELPSEERG	Sucrose binding protein homolog S-64

## Data Availability

All data generated or analyzed during this study are included in this published article and can be reused only with the authors’ permission.

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
