# Peer review of "Soybean-Derived Peptides Attenuate Hyperlipidemia by Regulating Trans-Intestinal Cholesterol Excretion and Bile Acid Synthesis"

_nutrients, 2021, doi:10.3390/nu14010095_

Round 1
Reviewer 1 Report
This article focuses on the cholesterol lowering effect of soybean peptides and found novel responsible peptides. As molecular mechanism, authors tried to show their promoting effect of TICE and cholesterol secretion into feces. Totally, the experimental design is appropriate and this article include novel findings in terms of nutritional point of view. On the other hand, there are several concerns in data presentation and obscurity in data interpretation to be revised or clarified as follows.
Major
- P5, L198; t-test is not suitable for the evaluation of this experimental design. Appropriate statistical method should be used.
- 1A; SDS-PAGE image is unclear. Although author explains “There was not detected for negative control…. P5, L210-211, the electrophoresis is not clear. It looks overload of protein.
- 1B; Soy protein upregulated ABCG5, 8. Please discuss the difference between protein and peptide. Does soy protein also activate LXR?
- 2; Author selected 1 ug/mL as concentration of peptide. Please explain the verification or how to decide this concentration. As shown in Fig. 2A, authors detected many signals in HPLC analysis. Please explain the reason why authors select only 3 peaks.
- 3C, 3D, 3E, 4D; These data do not show the effect of #1 and #8, therefore these data could not show the effect of GSK2033 on peptides.
- P9, L306-307; These statements can be misleading. As Cyp7A1, Cyp8B1 involve in bile acid synthesis from cholesterol, it appears that these enzymes are responsible for cholesterol synthesis. These results appear to be inconsistent when compared with serum cholesterol, how do you explain. In addition, hepatic cholesterol level and effect of these peptides on cholesterol synthesis such as HMGCR, SREBP should be shown or cited and discussed.
Minor
- P2, L114; What’s soy solution? How to prepare it and is it different from soybean protein?
- P4, L130; “whole” is duplicated.
- P4, L152; Author cited Reference 28 but differentiation period seems to 21 days in original whereas 7 days in this article. Original article used Caco-2 clone TC7. Please explain the validation of the culture protocol used in this experiment.
- P4, L164; Please show the wavelength for the detection in HPLC.
- P5, L188; Please clarify the solvent for the administration of peptides.
- P5, L188; Please clarify the way of anesthesia.
- P8, L287; Reference number is missing.
- 4; To verify siRNA knockdown FGF production from KD cells should be shown. I think this is supplement Figure.
- 5; How to distinguish proximal and distal intestine? Please show the criteria in the method.
- 5B; Please show the daily weight of feces to show substantial daily cholesterol excretion.
- 3, 5; Please quantify the western blot data and perform statistical analysis.
Reviewer 2 Report
The study of Lee et al. aims to evaluate the contribution of Soybean-derived peptides to hypercholesterolemia and propose mechanisms that only involve regulation of TICE and bile acid synthesis, but the pieces of evidence are weak, and more exhaustive analysis of cholesterol homeostasis pathways must be evaluated as well as TICE in vivo and bile acids analysis.
Although the TICE is extrapolated from in vitro studies, their finding needs to be validated in vivo (TICE assay after infusion of labeled cholesterol). This assay would definitively support their hypothesis, as many other mechanisms may be involved (Intestinal absorption and liver synthesis). Additionally, the authors should perform a cholesterol absorption assay to help evaluate de NPC1L1 pathways. NPC1L1 expression should be evaluated in Caco cells. HMGCoaR expression should be assessed in MIHA cells and livers.
Major comments:
- Based on total circulation cholesterol levels (displayed in figure 1E), I can conclude that soy hydrolysate has a significant pro-cholesterolemic effect almost similar to HCD. The Authors should clarify this fundamental issue.
- From a methodological point of view, their findings are highly questionable if mice do not gain weight upon HCD and peptides treatment (figure 5A) – we can speculate that the animals are unhealthy. Food intake and adiposity should be evaluated as lower cholesterol levels may result from lower HCD intake and peptide-mediated central effect. Cells toxicity of both peptides should also be assessed in vitro.
Overall, the manuscript kinetics and incubation periods of in vitro and in vivo experiments are not indicated.
Did the Authors evaluate the dose-dependency of peptides' effects? This may support the specificity of their findings.
The rationale for choosing the experimental doses of peptides should be discussed and argued.
Bile acids composition and their hydrophobic index may also be evaluated as they can contribute to cholesterol absorption.
Line 381, "which induced hepatic bile acid synthesis to support hepatobiliary cholesterol excretion, "It is an opposite conclusion compared with their results (a decreased expression of CYP7A1 and CYP8B1).
In all figures with western blot, there are no statistical pictures, it should be shown.
Round 2
Reviewer 1 Report
Authors have appropriately responded to all comments and questions.
Author Response
We appreciate the reviewer's comments.
Reviewer 2 Report
My perception of the results is that Soybean-Derived Peptides modulate trans-intestinal cholesterol excretion in vitro and bile acid synthesis in vivo, potentially contributing to lower cholesterol levels. The analysis of TICE in vivo is mandatory to keep the current Author’s title, indeed decreased cholesterol levels may be related to other pathways. Or the title should be adapted, and the limitations of the study should be discussed.
The argumentation for the lack of toxicity is not scientific. In vivo or in vitro toxicity tests are mandatory.
